# Fast amplitude modulation up to 1.5 GHz of mid-IR free-space beams at room-temperature

Stefano Pirotta[1✉], Ngoc-Linh Tran [1], Arnaud Jollivet[1], Giorgio Biasiol [2], Paul Crozat[1], Jean-Michel Manceau[1], Adel Bousseksou[1] & Raffaele Colombelli [1✉]

Applications relying on mid-infrared radiation ($\lambda \sim$ 3-30 μm) have progressed at a very rapid pace in recent years, stimulated by scientific and technological breakthroughs like mid-infrared cameras and quantum cascade lasers. On the other side, standalone and broadband devices allowing control of the beam amplitude and/or phase at ultra-fast rates (GHz or more) are still missing. Here we show a free-space amplitude modulator for mid-infrared radiation ($\lambda \sim$ 10 μm) that can operate at room temperature up to at least 1.5 GHz (−3dB cutoff at ~750 MHz). The device relies on a semiconductor heterostructure enclosed in a judiciously designed metal–metal optical resonator. At zero bias, it operates in the strong light-matter coupling regime up to 300 K. By applying an appropriate bias, the device transitions towards the weak-coupling regime. The large change in reflectance is exploited to modulate the intensity of a mid-infrared continuous-wave laser up to 1.5 GHz.

[1] Centre de Nanosciences et de Nanotechnologies (C2N), CNRS UMR 9001, Université Paris-Sud, Université Paris-Saclay, 91120 Palaiseau, France.
[2] Laboratorio TASC, CNR-IOM, Area Science Park, 34149 Basovizza, Trieste, Italy. ✉email: stefano.pirotta@u-psud.fr; raffaele.colombelli@u-psud.fr

Fast amplitude and phase modulation are essential for a plethora of applications in mid-infrared (IR) photonics, including laser amplitude/frequency stabilization[1], coherent detection, FM (frequency modulation) and AM (amplitude modulation) spectroscopy and sensing, mode-locking, and optical communications[2,3]. However, the fast and ultra-fast (1–40 GHz) modulation of mid-IR radiation is a largely under-developed functionality. The fastest modulation speeds, 20–30 GHz, have been obtained with the direct modulation of mid-IR quantum cascade lasers (QCLs), but this requires specially designed devices and elevated injected RF (radiofrequency) powers[4–6]. Interestingly, in the visible/near-IR spectral ranges the preferred solution is to separate the functionalities: independent modulators, filters, interferometers are employed that are physically separated from the source. For modulators, this leads to advantages in terms of RF power, laser linewidth and flatness of the modulation bandwidth.

Commercially available mid-IR modulators are either acousto-optic devices with narrow modulation bandwidth, or very narrow band ( ~100 kHz) electro-optic modulators based on GaAs or CdTe[7]. The latter ones can operate up to modulation speeds of 20 GHz, but their efficiency is very low: <0.1% sideband/carrier ratio (see "Methods" for definition). To date, standalone, efficient and broadband amplitude/phase modulators are missing from present mid-IR photonics tools. Holmström[8] shows numerical performances up to 190 GHz with step QWs in a waveguide geometry at $\lambda = 6.6\,\mu m$, but no experimental data are provided.

Since the 80s, proposals have been put forward to exploit intersubband (ISB) absorption in semiconductor quantum well (QW) systems to modulate mid-IR radiation. The first attempts based on the Stark shift were then followed by a number of works exploiting coupled QWs[9,10]. In both cases, the application of an external bias depletes or populates the ground state of the QW at cryogenic temperatures (from 4 K up to 130 K) thus inducing a modulation of the ISB absorption[11]. Operation at room-temperature was obtained in ref. [12] using a Schottky contact scheme. Recently, different approaches have been proposed to actively tune the reflectance/transmission of mid-IR and/or THz beams: phase transition in materials like $VO_2$, liquid crystals orientation, carrier density control in metal–insulator–semi-conductor junctions[13]. These devices operate on the principle that, at a given wavelength, a change in absorption translates in a modulation of the transmitted power. An alternative approach is to frequency shift the ISB absorption, instead of modulating its intensity. In[14–17] a giant confined Stark effect in a coupled QW system embedded in a metallic resonator, designed to be in the strong coupling regime between light and matter, was exploited. A response time of ~10 ns was estimated. Exploiting the Stark effect in ISB-based systems can effectively lead to impressive performances, but it can suffer of an intrinsic drawback: the diagonal transition has lower oscillator strength with respect to a vertical one. Higher doping is necessary to achieve the same Rabi splitting: this means higher biases to get a frequency tunability comparable to that obtained in systems based on charge modulation. On the other hand, high biases can be a significant problem when targeting fast and/or ultra-fast performances.

One way forward is photonic integration: scaling to the mid-IR the approach already developed for silicon photonics. It can rely on SiGe/Si photonic platforms[18], or on the more natural InGaAs/AlInAs-on-InP platform[19]. In both cases the QCL source must be properly integrated in the system. An alternative is to develop modulators that can apply an ultra-fast RF modulation to a propagating beam, either in reflection or in transmission. This approach does not require a specific integration of the source and can in principle be applied to laser sources beyond QCLs.

In this article we follow the latter strategy by proposing a standalone device capable of modulating a mid-IR beam at room-temperature up to 1.5 GHz. It's based on a GaAs/AlGaAs heterostructure, embedded in a metal–metal optical resonator (scheme in Fig. 1(a)). The system is designed to operate in

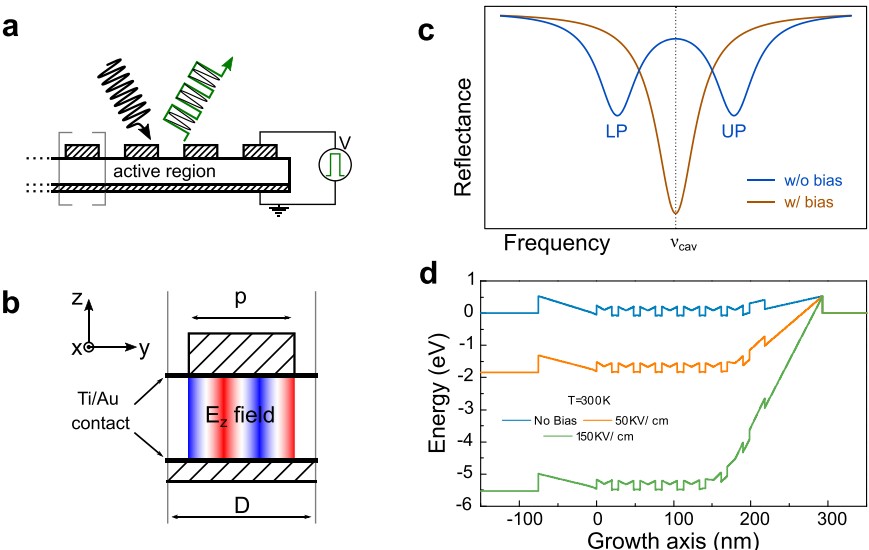

**Fig. 1 Modulation operating principle. a** Sketch of the modulator geometry: the active region is embedded in a metal–metal structure. By applying on it an external bias, the amplitude of the reflected beam is modulated. **b** Close-up of a single 1D ribbon of the device. p is the patch side, D the period. Two Schottky contacts permit the effective application of an external bias to the hetero- structure. The electric-field distribution (z-component) of the $TM_{03}$ mode is also sketched. $TM_{0i}$ refers to the mode with i nodal lines in the y-direction. **c** Intuitive view of the modulator operating principle in an ideal configuration: with no applied bias the system is designed to be in strong coupling. Two polaritonic branches are visible in the reflectance spectrum, for a specific value of the patch side p. By applying a specific bias we can deplete an arbitrary number of quantum wells and bring the system in the weak-coupling regime: the bare cavity $TM_{03}$ mode is visible on the reflectance spectrum. **d** Conduction band profile obtained by solving the Schrodinger–Poisson equation (NextNano++[24]) at room-temperature with increasing external bias. No bias (solid blue line), 1.84 V (orange solid line) and 5.52 V (green solid line).

reflectance: it is in strong coupling when no external bias is applied. We demonstrate a clear modulation of the strong coupling condition after bias application. The response bandwidth has been measured at room-temperature, showing a −3 dB cutoff at 750 MHz and the modulation of a mid-IR laser beam up to 1.5 GHz has also been reported.

## Results

**Strong coupling modulation leads to reflected beam modulation.** Our approach is to operate the device in the strong light-matter coupling regime, and introduce the fast modulation by— ideally— switching the system in and out of strong coupling with the application of a bias voltage. A periodic QW structure is embedded in an optical resonator composed by non-dispersive metal–metal one-dimensional (1D) ribbons (or 1D patch cavities[20]) as shown in Fig. 1(a). The system, designed to operate in reflectance, is conveniently optimized so that the ISB transition is strongly coupled to the $TM_{03}$ photonic mode of the resonator, whose electric-field distribution is shown in Fig. 1(b): the notation convention $TM_{0i}$ is defined in the caption. The resulting reflectance $R_{NB}$ (NoBias reflectance) is sketched in Fig. 1(c), solid blue line. Let's consider a laser tuned to the bare cavity frequency $v_{las} = v_{cav}$ impinging on the device: almost all the intensity is reflected back since $R_{NB}(v_{las}) \sim 1$. By applying a bias to the structure we can effectively empty an arbitrary number of wells and finally change the coupling condition between the cavity mode and the ISB transition. In the best case scenario—the ideal device—we can induce a transition to the weak-coupling regime. The corresponding reflectance spectrum is sketched in Fig. 1(c) (solid orange line): only the bare cavity transition is visible, as polaritons are no more the eigenstates of the system. At the laser frequency $v_{las} = v_{cav}$ we have $R_B(v_{las}) \ll R_{NB}(v_{las})$, being $R_B$ the reflectance under bias B: the reflected laser beam is amplitude modulated with an elevated contrast. Contrast and modulation height are used as synonyms in the paper: the use of the one or the other is only dictated by text readabilty (see Reflectance under DC external bias section below and Fig. 2 for its quantitative definition).

As the intersubband polariton dynamics features ps-level timescales[21], the bandwidth of the modulator is limited by (i) the RC-constant of the circuit and (ii) the transfer time of electrons in/out of the QWs. In fact, the top and bottom metal-semiconductor Schottky interfaces permit the application of a gate to the multiple QW structure that can efficiently deplete the system, as shown in Fig. 1(d). Note: the electrical control of ISB polaritons, in a quasi-DC regime though, has been studied in refs. [22,23].

We highlight the importance of the strong coupling regime between the ISB transition and the patch cavity mode to achieve an effective free-space modulation of mid-IR laser beams, in particular as far as the spectral agility is concerned[14,15]. By optimizing the design it is possible to obtain amplitude modulation over a broad range, and a significant contrast even at 300 K operating temperature. On the other hand, operating the device in the weak-coupling regime (in a metal–metal cavity for instance) would indeed behave differently. Modulating the absorption, or even tuning the frequency of the ISB transition, would just mildly affect the resonance linewidth: the resulting modulation range and contrasts would be much smaller.

**Sample fabrication.** The semiconductor heterostructure was grown by solid-source molecular beam epitaxy on an un-doped GaAs substrate. It is composed of seven periods of 8.3 nm GaAs QWs separated by 20 nm-thick $Al_{0.33}Ga_{0.67}As$ barriers. Si delta-doping ($n_{Si} = 1.74 \times 10^{12}\ cm^{-2}$) is introduced in the barrier

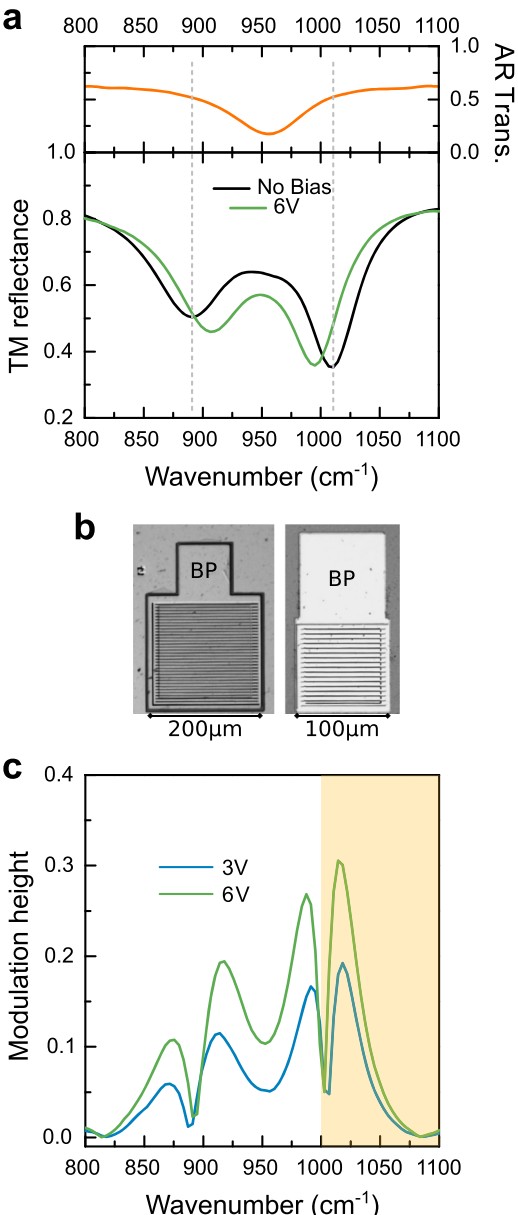

**Fig. 2 Active region characterization and device reflectance under DC bias. a** Top panel: transmission through the bare active region in a typical multipass geometry, at room-temperature: a clear ISB transition at about 955 cm⁻¹ is measured. Bottom panel: room-temperature reflectance spectra under FTIR-coupled microscope of the $p = 4.2\ \mu m$ sample. The black solid line is the reflectance at no bias; the green solid line corresponds to the application of +6 V. The Rabi-splitting decreases by 25%. **b** Optical microscope images of the large devices (left; surface $5 \times 10^4\ \mu m^2$) and of the small ones (right; surface $2 \times 10^4\ \mu m^2$). The bonding pad (BP in the figure) has the same dimensions ($100 \times 100\ \mu m^2$) in both samples. **c** Modulation height extracted from the spectra of panel **a** (large devices). It is defined as $\min\left(\left|\frac{R_{NB}-R_B}{R_B}\right|, \left|\frac{R_{NB}-R_B}{R_{NB}}\right|\right)$ and it is plotted in the range 800–1100 cm⁻¹ for both 3 V (blue solid line) and 6 V (green solid line) biases. The semi-transparent orange region corresponds to the nominal wavelength coverage of our commercial, tunable mid-IR QC laser source.

center. A 40 nm-thick GaAs cap layer terminates the structure, and a 500 nm-thick $Al_{0.50}Ga_{0.50}As$ layer is introduced before the active region, whose total thickness is $L_{AR} = 368.1$ nm. The sample presents an ISB transition at an energy of 118.5 meV (about 955.8 cm⁻¹), that we have measured at 300 K in a classic

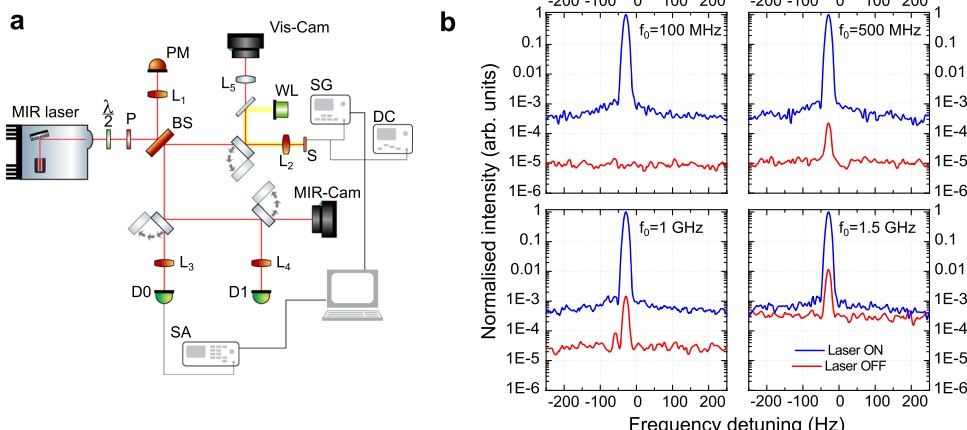

**Fig. 3 Polariton-based amplitude modulation up to 1.5 GHz at 300 K. a** Sketch of the experimental setup to measure the modulator bandwidth. The sample S is pumped with a commercial tunable mid-infrared QC laser focused with a ZnSe ($L_2$) lens; the back reflected beam is collected through the beam-splitter BS, and sent to the detectors $D_0$ (fast MCT Vigo detector with nominal bandwidth 1 KHz–837 MHz) and $D_1$ (general purpose 50 MHz-bandwidth MCT detector, LN-cooled). A signal generator (RF or LF depending on the measurement) is used to apply the electrical bias to the sample. The electrical signal from the detector is then sent to specific analyzers: a spectrum analyzer (SA) to collect the beat-note spectrum or a lock-in. PM power meter, WL white light, P polarizer, $\lambda/2$ half-wave plate, Vis-Cam visible camera, MIR-Cam MIR camera. **b** Normalized beat-note spectra obtained when the sample is fed with 10 dBm/DC@−989 mV with modulation frequencies 100 MHz, 500 MHz, 1 GHz, and 1.5 GHz from top left. The measurements are performed at room-temperature, and the measured sample is the small one (surface is $2 \times 10^4\ \mu m^2$, $p = 4.1\ \mu m$). The QC laser frequency is 1010 cm$^{-1}$. Two curves are shown for each panel: laser-on (solid blue line) and laser-off (solid red line). The modulator performs up to at least 1.5 GHz.

multipass waveguide transmission configuration (orange spectrum in Fig. 2(a)). Figure 1(d) shows the global conduction band profile at room-temperature (RT, solid lines) and at different applied biases for the fabricated structure. It was obtained solving self-consistently the Schrodinger–Poisson equations using a commercial software[24]. With no applied bias all the QWs are populated. The application of a bias gradually depletes them.

The modulators rely on a metal-semiconductor-metal geometry. We have wafer-bonded the sample on a n$^+$-GaAs carrier layer via Au-Au thermo-compression wafer-bonding, a standard technology for mid-IR polaritonic devices[25,26]. After polishing and substrate removal, the 1D patches are defined with electron-beam lithography followed by Ti/Au deposition (5/80 nm) and lift-off. The top contact patterning and the definition of the bonding pads are realized with optical contact lithography and Ti/Au lift-off. An inductively coupled plasma (ICP) etching step down to the back metal plane defines the mesa structure. Optical microscope images of typical final devices are shown in Fig. 2(b). Arrays of devices have been fabricated that differ in the width p of the metallic fingers (nomenclature in Fig. 1(b)). For each value of p, we fabricated two arrays with different total surface ($5 \times 10^4$ and $2 \times 10^4\ \mu m^2$, respectively. Fig. 2(b)). The active region being very thin (368.1 nm), the system does not operate as a photonic-crystal, but operates instead in the independent resonator regime. The cavity resonant frequency $\nu_{cav}$ is set by $p$, not by the period $D$, according to the following expression:

$$\frac{c}{\nu} = \lambda = \frac{2\ n_{eff}\ p}{i} \quad \text{where } i \in \mathbb{N}. \tag{1}$$

The system behaves as a Fabry–Perot cavity of length $p$, with $n_{eff}$ an effective index that takes into account the reflectivity phase at the metallic boundaries[20,27]. We opted to operate not on the $i = 1$ fundamental mode, the standard choice[20], but on the $i = 3$ mode (the TM$_{03}$), to simplify the fabrication procedure and increase the electromagnetic overlap factor. Supplementary Figs. 1 and 2 provide a justification for this choice.

**Reflectance under DC external bias.** The reflectance of the devices as a function of p has been measured with a microscope

coupled to a Fourier transform infrared spectrometer (FTIR) to retrieve the polaritonic positions at RT (and at 78 K) with no applied bias. The complete dispersion is shown in Supplementary Fig. 2: together with the measurements and simulations on an empty cavity (Supplementary Fig. 1), it permits to identify the first 3 ribbon resonator modes. The TM$_{03}$ mode exhibits a clear Rabi splitting for patch sizes around $p = 4\ \mu m$.

A suitable device ($p = 4.2\ \mu m$) was wire bonded and its reflectance was measured under different applied DC biases. When no bias is applied, we observe the two polariton branches (black curve in Fig. 2(a)). The green solid line corresponds to a +6 V bias, that is practically the limit imposed by the Ti/Au Schottky barrier. A very similar behavior is observed for negative biases given the symmetry of the top and bottom contacts' barriers. The Rabi splitting decreases by 25%: it means that the gate empties only half of the QWs, as $\Omega_{Rabi} \propto \sqrt{N_{QW}}$, being $N_{QW}$ the total number of QWs in the structure. A second sample with lower doping ($N_{Si} = 6.2 \times 10^{11}$ cm$^{-2}$) has been characterized. This sample can fully transition to the weak-coupling regime upon application of a bias (see Supplementary Fig. 3). However, it's inferior to the highly doped one in terms of modulation height given the wavelength coverage of our tunable laser. For this reason we preferred to work with the highly doped device.

From the measurements we can extract the modulation height attainable on an incoming laser beam with a +6 V maximum bias with this specific device. The modulation height is defined as $\min\left(\left|\frac{R_{NB}-R_B}{R_B}\right|, \left|\frac{R_{NB}-R_B}{R_{NB}}\right|\right)$: it is plotted in Fig. 2(c) in the 800–1100 cm$^{-1}$ range for both $B = 6$ V and $B = 3$ V. It shows that a contrast above 10% can be obtained in a few frequency ranges. In particular, a contrast between 20% and 30% can be obtained around 1030 cm$^{-1}$ ($\lambda \sim 9.70\ \mu m$). This frequency is covered by our tunable commercial QC laser (shadowed orange region in Fig. 2(c)).

**Reflectance modulation up to 1.5 GHz.** We have measured the speed and modulation bandwidth of the modulator with the setup described in Fig. 3(a). A continuous-wave (CW), tunable commercial QC laser[28] is focused on the modulator (S) that is fed

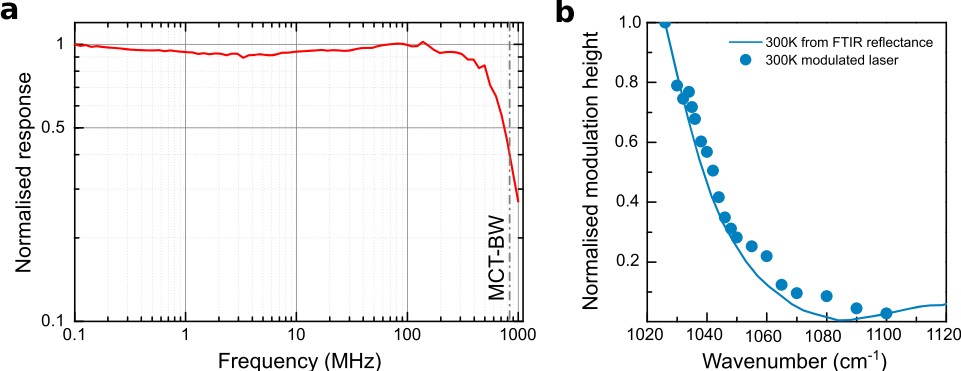

**Fig. 4 Polariton-based amplitude modulator bandwidth at 300 K. a** Normalized response at room-temperature of the small device ($p = 4.1\,\mu m$) when fed with 10 dBm/DC@−989mV. The QC laser frequency is 1010 cm$^{-1}$. The nominal −3 dB cutoff at 837 MHz of the fast MCT is shown as a gray dashed vertical line. The entire experiment is automated by an homemade python code, based on the Pymeasure package (https://pymeasure.readthedocs.io/en/latest/). **b** Normalized modulation height as a function of the QC laser frequency (dots). The measurements are performed on the large devices (surface is $5 \times 10^4\,\mu m^2$, $p = 4.2\,\mu m$). The QC laser frequency is tuned between 1026 and 1100 cm$^{-1}$ and the modulated beam is detected by the 50 MHz MCT and the lock-in. The modulator is fed with a 0–3 V sinusoidal signal at 100 kHz. The solid line is the modulation height for 3 V bias of Fig. 2(c), normalized to 1 at 1026 cm$^{-1}$. The perfect overlap proves that the origin of the modulation is indeed the transition from the strong light-matter coupling regime towards the weak-coupling.

with an RF signal from a synthesizer (SG). The reflected, and modulated, beam is detected with a 837 MHz-bandwidth commercial MCT detector (D0,[29]) whose output is fed to a spectrum analyzer (SA); or—for low-frequency measurements—to a 50 MHz-bandwidth MCT detector (D1) and then to a 200 MHz lock-in amplifier. Visible and mid-infrared (MIR) cameras (Vis-Cam and MIR-Cam in Fig. 3(a)) permit a correct beam alignment on the sample, with the help of an external white light source (WL). All the measurements are performed at room-temperature (300 K).

Figure 3(b) shows the spectra obtained from the smaller modulator ($p = 4.1\,\mu m$), using a QCL frequency of 1010 cm$^{-1}$. The sample is driven with an RF signal of constant power (DC offset −989 mV), but different frequency (100 MHz, 500 Mhz, 1 GHz, and 1.5 GHz). The reflected beam is detected with a fast MCT, whose signal feeds the SA, in both laser-on (blue solid line) and laser-off (red solid line) configurations. In the latter case, the presence of a peak on the noise floor is due to some direct cross-talk between the RF synthesizer and the spectrum analyzer through the RF injection and detection circuits. The normalization to 1 allows the comparison at different frequencies. We can detect a signal up to a modulation speed of 1.5 GHz, well beyond the VIGO detector 3 dB cutoff of 837 MHz, proving the fast character of our modulator.

In order to determine the modulator bandwidth, we performed an automated scan as a function of the modulation frequency. It consists in acquiring the beat-notes (as in Fig. 3(b)) at closely spaced frequencies between 0.1 MHz and 1 GHz. The software acquires the peak amplitude with noise floor correction at each frequency, and the data are normalized to 1 at the lowest RF frequency of the scan (0.1 MHz). The results, at 300 K, are reported in Fig. 4(a) for a typical $2 \times 10^4\,\mu m^2$ device, with grating period $p = 4.1\,\mu m$. At the optimum performance point ($\nu_{laser} = 1010$ cm$^{-1}$), it operates at frequencies > 1 GHz, with a −3 dB cutoff at ~750 MHz (Fig. 4(a), red curve). The larger devices (data not shown) typically exhibit a −3 dB cutoff at ~150 MHz. This result is in fair agreement with the surface ratio between the two devices. Furthermore the theoretical RC-cutoff of the large samples is $f_{cutoff}^{large} = \frac{1}{2\pi RC} = 204$ MHz and for the small sample $f_{cutoff}^{small} = 510$ MHz ($C$ device capacitance and $R$ the 50 $\Omega$ output resistance of the RF synthesizer). The good agreement proves that the bandwidth is currently limited by the RC time constant. For

high-resolution spectroscopy, an important parameter is the sideband/carrier power ratio. For the current modulators, from quasi-DC response measurements (not shown) we estimate a ratio of the order of 5%.

If the QC laser frequency is tuned away from the optimum value, Fig. 2(c) predicts that the modulation should drop. This observation is crucial to unambiguously assign to the polariton modulation the enabling physical principle of the device. To this scope we have measured, at room-temperature, the modulation height as a function of the QCL laser frequency. The results—normalized to 1 —are reported in Fig. 4(b) (dots) for the larger sample ($p = 4.2\,\mu m$, the same of Fig. 2(c)), and they are superimposed to the DC modulation response curve obtained from the reflectance measurements from Fig. 2(c). The modulator fast response as a function of the impinging laser frequency closely follows the DC contrast curve. This finding confirms that the enabling mechanism is indeed the fast modulation of the Rabi splitting via application of an RF signal.

## Discussion

Having established that the current devices are RC limited, the natural question is: what is their intrinsic speed? The physics of the current devices is not very different from the one of mid-IR quantum well infrared photodetectors (QWIP), except the absence of the ohmic contacts, that are known to operate up to speeds of 60/80 GHz[30,31]. That alone suggests that the intrinsic speed of the current modulators is set by the same parameters, in particular the capture time $\tau_{cap}$ and the transit time $\tau_{trans}$ that is set by the drift velocity. There is, however, a notable difference: in the ideal operating regime, carriers diffuse all the way towards one metal-semiconductor interface upon application of a bias. Moreover, they have to flow back through the active region when the bias is restored to 0. This leads of a characteristic time $\tau_{drift} = \frac{L_{AR}}{v_{drift}}$. In our case, with a very conservative $v_{drift} = 10^6\,\frac{cm}{s}$ at RT[32], we obtain a value of $\tau_{drift} < 30$ ps, which sets a lower bound for the intrinsic cutoff in the range 5–10 GHz ($\frac{1}{2\pi\tau}$ estimation).

In conclusion, we have demonstrated a technology that is able to amplitude modulate mid-IR free-space laser beams up to GHz modulation frequencies. In this first demonstration, at $\lambda = 9.7$ μm, we achieved modulation speeds up to 1.5 GHz at room-temperature (−3 dB cutoff at ~750 MHz). The device operates by modulating the strong coupling regime at fast rates, one of the

few demonstrations of a practical device relying on the strong light-matter coupling regime. The estimated intrinsic speed is at minimum 5 GHz. Improved active regions that do not rely on drift transport, but instead on tunnel coupling[23] will probably lead to modulation speeds in the 30/40 GHz range.

## Methods

**Sideband to carrier ratio**. In amplitude modulation (AM), it gives the quality of the signal modulation. In the most simple situation where both carrier and signal are sinusoidal, the carrier is $c(t) = C \sin(2\pi\nu_c t)$ while the signal can be written as follows: $s(t) = S \cos(2\pi\nu_s t + \phi) = Cm \cos(2\pi\nu_s t + \phi)$. We have defined the modulation index $m = \frac{S}{C}$: it measures how deep the carrier modulation is, with $m = 1$ corresponding to 100% modulation. In the frequency domain, the carrier line ($C$ intensity) and the two sidebands at $\nu_c \pm \nu_s$ with amplitude $\frac{1}{2}Cm$ appear. The sideband to carrier ratio is then $m/2$.

## Data availability

All relevant data are available from the authors upon reasonable request.

## Code availability

Python code is also available from the authors upon reasonable request.

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

## Acknowledgements

We thank S. Barbieri, J.-F. Lampin and E. Peytavit for useful discussions. We also thank L. Wojsvzyk, A. Nguyen, and J.-J. Greffet for the loan of the 50 MHz-bandwidth MCT detector. We acknowledge financial support from the European Union FET-Open Grant MIRBOSE (737017). This work was partly supported by the French RENATECH network. R.C. and A.B. acknowledge financial support from the French National Research Agency (project "IRENA").

## Author contributions

G.B. grew the sample; N.-L.T. fabricated the sample, performed measurements and simulations; R.C. designed the devices, performed simulations, and supervised the entire project; P.C. helped in RF setup; S.P. performed simulations, built the RF setup and performed the measurements; A.J. performed simulations. All the authors (S.P., R.C., N.-L.T., A.J., G.B., P.C., J-M.M., A.B.) discussed data and wrote the manuscript.

## Competing interests

The authors declare no competing interests.
