## [Peer Review File · Nature Communications]

Reviewers' Comments:

Reviewer #1:

Remarks to the Author:

The Authors report on an intersubband-based high speed modulator for Mid-IR free space beam operating at room temperature. The device operate up to 1.5 GHz at a tested optical frequency of approximately 30 THz.

The device concept is new and the results are surely interesting. The paper is generally well written apart from some scattered typos and some points that in my opinion need clarification. I list here below my concerns/questions:

-The key concept and innovation that the Authors claim is the use of strong light matter coupling to enhance the spectral agility of the device with respect to the older ISB realisations based on Stark shift. In terms of the fundamental speed limits it is not entirely clear to me the role played by the "polaritonic" nature of the device with respect to a more standard ISB absorption. The ps dynamics is not limited to polaritons but is a general feature of intersubband transitions. Do the Authors think that a QW in the weak coupling regime in a similar, small double-metal cavity would behave differently ? They even say themselves (page 6) that "The physics of the current devices is not very different from the one of mid-IR QWIP "

Doesn't it all boils down to moving the carriers in-and-out of the wells to change absorption (reflection) profile? In my opinion the paper does not underline enough the peculiarity and the necessity to achieve strong-light matter coupling to target the desired performance.

Figure 3(b): In this figure I have some troubles understanding the reported quantities:

-what does it mean to have a signal even when the laser is off? is this emission from the modulator due to carriers moving around?

-Do the Authors have an idea of the Mid-IR spectrum of this signal at the modulation frequency with the laser off (red trace)

-paragraph 2 is entitled "switching between strong and weak coupling regimes" but the reported data always shows two nice and well separated polaritonic peaks and the actual light-matter coupling decreases by 25%, not by 100%. I am not sure that is the appropriate title . I would more go for "modulation of the strong coupling regime"

-the terminology "ultra-fast" for an object with a cutoff slightly above 1 GHz is in my opinion abused. Ultrafast physics (few optical cycles) happens on timescales well below 200 fs for visible radiation . for Mid-IR this would translate into few ps dynamics, not nanosecond or slightly below....I would suggest to replace the term with something more descriptive of the actual device performance.

Minor/technical remarks

-It would be worth citing, in the same spirit of Ref.15, also Jessop et al., Applied Physics Letters, 108 (2016)

-Figure 1 (b) would benefit from a x-y axis schematic

-Figure 2(c): wouldn't be better to report also the contrast, as this is the quantity mainly discussed in the text?

-Figure 4 (a): I would use a dB vertical scale, so the -3 dB cutoff would be immediately evident.

-The modulator operates in reflection geometry, that for some application is not ideal. Do the Authors have prospects for implementing such concepts in a transmissive device?

- I have a suggestion for an experiment that could shed more light into the modulator's behaviour:

the reflected/modulated signal could be filtered with an FTIR and then detected with a (fast) lock-in, in the same spirit as the SWIFTS spectroscopy done on QCL combs. In this way it could be assessed the Mid-IR spectral content that is modulated. I understand that it is a quite involved measurement and it is not strictly required for the present paper but it could be valuable for future works.

To conclude, the paper cannot be accepted in the present form as the Authors need to clarify the aforementioned points

Reviewer #2:

Remarks to the Author:

This paper demonstrates a mid-infrared intensity modulator for modulating beams in free space at room temperature, with an operation speed up to 1.5 GHz. The device consists of a QW structure embedded in an optical resonator, and it modulates the light by switching the system in and out of the strong coupling regime. The results presented show a breakthrough in terms of speed compared to the reported works. However, it is difficult to be considered as a significant achievement. The working principle is similar to Refs 15 and 16. The major difference is that the way of tuning the ISB transitions is replaced by depleting the QWs using a Schottky contact, which is applied in Ref. 13. Based on this I also disagree with the claim in the conclusion that this is possibly the first report of practical device relying on the strong light-matter coupling. Because of the similar principle, the breakthrough in the speed is mainly contributed by the smaller device dimension that reduces the RC time constant. E.g. the larger device in this work with a surface area of $5 \times 10^4 \mu\text{m}^2$, has a cut off frequency at 150 MHz. Ref 16 achieves a 10 ns response time (pulsar source limited) with a $16 \times 10^4 \mu\text{m}^2$ surface area, which is comparable. Therefore, I do not recommend it for publication in Nature Communications.

Besides, I would like to point out some issues I found in the paper:

1. The paper is mostly well written, but overall it is so concise that necessary details are missing to provide a comprehensive understanding. For example:
 - a. Section 2 is too short to clearly introduce the working principle of the device.
 - b. Insets of Fig. 1(c) and the symbols are not detailed.
 - c. There are too little comments on Fig. 3(b), making it difficult to understand what the data represent
 - d. There's no definition for the 'normalized response' in Fig. 4(a)
 - e. No introduction to 'sideband/carrier ratio'
2. The theoretical Rabi splitting example shown in Fig. 1(c) is too ideal compared to the practical device. The two polaritons are too separated to leave a long region of $R \sim 1$ (may refer to Fig.1 in DOI: 10.1039/c7nr06917k). In the introduction, it is better to be introduced as a significant drop on the reflectance when switching from the strong coupling regime to the weak coupling regime, rather than saying $R=1$ to $R=0$, as we only have 10% change in the experiment. It would also be better to change the x-axis to frequency to be consistent with the descriptions in the caption and the main text.
3. Both Fig. 1(c) and Fig. 2(a) are better to be plotted without an offset to directly observe the reflectance change. +6V and -6V of Fig. 2(a) can be plotted separately but compared with the zero bias. Meanwhile, the reflectance values are important for Fig. 2(a), which provides the information of how much the reflectance change and the insertion loss. These modifications also facilitate correlating them to Fig.2(c).
4. It's better to label the surface area or the length of each boundary in Fig. 2(b), pointing out

which is the bonding pad and which is the wire gating area.

5. γ -label of Fig. 2(c) can be changed to modulation depth since it has been defined in the main text. Too difficult to read the equation in the label.

6. The functions of some components in Fig. 5(a) are not introduced, e.g. PM, Vis-Cam, WL and MIR Cam. If they are unnecessary to be introduced they can be removed from the diagram. Otherwise some simple introductions are needed.

7. Except the speed difference, are there any differences in the performance between the smaller and the larger devices?

8. Some typos:

'Ghz', and the missing of the frequency number (The QC laser frequency is ...) in the caption of Fig. 3.

The paragraph below Fig. 4, 'a typical 100x100 μm^2 '. Should it be $2 \times 10^4 \mu\text{m}^2$?

Reviewer #3:

Remarks to the Author:

In this manuscript, the authors present a novel type of mid-infrared modulator surface based upon an intersubband polariton. While modulation of infrared light by intersubband transitions has been performed before numerous times, the use of a polaritonic transition to do so is relatively new, and no other report of modulation above 1 GHz has been reported. Hence the demonstration is novel, and the importance is potentially high, since the mid-infrared lacks for high-speed modulator components that are separate from a QCL. The results are convincing, particularly where the reflectance spectrum response measured by FTIR is matched with the modulation strength measured using a tunable laser.

Therefore, I recommend publication provided several points can be addressed :

- I am confused by the spectra in Fig. 3(b). It appears there is a modulated signal even when the mid-IR laser is not on? Can the authors explain this? Is it related to modulation of reflected background radiation by the modulator? If so, why is it not present for modulation at 100 MHz, but it is present for the other 3 modulation frequencies? Why is the noise background higher for when the laser is on?

- Given that Lee et al (Adv. Opt. Mat. 10.1002/adom.20140018) previously demonstrated a high-speed polaritonic mid-IR modulating metasurface, the statement in the conclusion that this is the "possibly the first report of a practical device relying on the strong light-matter coupling regime" is not warranted.

- What about modulation at voltages greater than 6 V? Can the polariton be completely switched off?

- In general the preparation and proofreading is a little sloppy.

o The Supplementary has many typos and misspellings and there appear to be missing references in it.

In the main text I noted:

o Fig 4. Caption – no units are given for area.

o Misspelling: "synthesizer"; "bandwidth", "Ghz", "stark" (should be capitalized),

o Italicization is haphazard – some variables are not italicized and should be, some units are italicized and should not be.

First of all, we want to thank all the three reviewers for the time devoted to read and comment our work. The following pages present our point-by-point answer to each one of the reviewer's comment.

Reviewer #1 (Remarks to the Author):

The Authors report on an intersubband-based high speed modulator for Mid-IR free space beam operating at room temperature. The device operate up to 1.5 GHz at a tested optical frequency of approximately 30 THz.

The device concept is new and the results are surely interesting. The paper is generally well written apart from some scattered typos and some points that in my opinion need clarification. I list here below my concerns/questions:

1 - The key concept and innovation that the Authors claim is the use of strong light matter coupling to enhance the spectral agility of the device with respect to the older ISB realizations based on Stark shift. In terms of the fundamental speed limits it is not entirely clear to me the role played by the “polaritonic” nature of the device with respect to a more standard ISB absorption. The ps dynamics is not limited to polaritons but is a general feature of intersubband transitions. Do the Authors think that a QW in the weak coupling regime in a similar, small double-metal cavity would behave differently ?

Indeed the main advantage of exploiting the strong coupling regime, and therefore the presence of polaritons, is the spectral agility: it permits to achieve modulation over a relatively broad spectral range as confirmed by Figure 2(c). And, in optimized structures, the modulation contrast can be very high. We also think that the beauty and the simplicity of the proposed device is to be taken into account.

To precisely answer the referee question, a single QW in the weak-coupling regime in a metal-metal cavity would indeed behave differently. Modulating the absorption would just mildly affect the resonance linewidth: the modulation range and contrasts would be much smaller.

The alternative is using a spatially indirect transition in a double QW, that can be tuned by Stark shift. On one hand the oscillator strength changes when applying a bias, complicating the design. On the other hand, we would have the same problem as in the single QW case: low contrast as we would have essentially a linewidth change, albeit larger than in the single QW case.

In terms of speed, the intrinsic dynamics is dominated by times on the order of ps for the ISB transitions in weak coupling as well as for ISB polaritons. If we focus on the geometry consisting of 2 coupled QWs, where a significant Stark effect exist (see for example Fig 1 in Ref. [15]), to our knowledge there is however very few literature on the high speed properties/capabilities of those systems. Holmström (reference added in the main text) shows numerical performances up to 190 GHz with step QW in a waveguide geometry at around 7 μm , but no experimental data are provided.

They even say themselves (page 6) that “The physics of the current devices is not very different from the one of mid-IR QWIP ”

Doesn't it all boils down to moving the carriers in-and-out of the wells to change absorption (reflection) profile? In my opinion the paper does not underline enough the peculiarity and the necessity to achieve strong-light matter coupling to target the desired performance.

We mentioned the mid-IR QWIPs to provide a path to better understand the detailed physics of the current devices. As we tried to convey in the paragraph above, it is the operation in strong coupling that provides improved functionalities in terms of spectral range and contrast, not the specific way we inject/extract charges.

Ultimately however the way we transfer charges in/out of the active QWs will become important for the speed. In the current geometry, the time carriers need to drift back to the emptied QW eventually limits the device speed. This limitation can be lifted by employing a different quantum design where a larger well acts like a reservoir for carriers that can tunnel (at a specific bias) through a barrier into a thinner, active QW.

In order to better clarify the advantages of the polaritonic approach, we added the following paragraph (page 3):

“We highlight the importance of the strong coupling regime between the ISB transition and the patch cavity mode to achieve an effective free-space modulation of mid-IR laser beams, in particular as far as the spectral agility is concerned [15,16]. By optimizing the design it is possible to obtain amplitude modulation over a broad range, and a significant contrast even at 300K operating temperature. On the other hand, operating the device in the weak-coupling regime (in a metal-metal cavity for instance) would indeed behave differently. Modulating the absorption, or even tuning the frequency of the ISB transition, would just mildly affect the resonance linewidth: the resulting modulation range and contrasts would be much smaller.”

2 - Figure 3(b): In this figure I have some troubles understanding the reported quantities:

-what does it mean to have a signal even when the laser is off? is this emission from the modulator due to carriers moving around?

-Do the Authors have an idea of the Mid-Ir spectrum of this signal at the modulation frequency with the laser off (red trace).

The residual signal we have on the spectrum analyzer (SA), even when the MIR laser is off, is pure RF noise coming from the measurement circuit. In fact we keep the RF synthesizer on during the measurement of the red curve with laser off, and there is evidently a small cross talk between the synthesizer and the spectrum analyzer. This often happens in RF setups. The red curves are essentially our noise floors.

We have expanded section 4 to better explain the results in Figure 3.b with this paragraph:

“The sample is driven with an RF signal of constant power (DC offset -989mV), but different frequency (100 MHz, 500 Mhz, 1 GHz and 1.5 GHz). The reflected beam is detected with a fast MCT, whose signal feeds the SA, in both laser-on (blue solid line) and laser-off (red solid line) configurations. In the latter case, the presence of a peak on the noise floor is due to

some direct cross-talk between the RF synthesizer and the spectrum analyzer through the RF injection and detection circuits.. The normalization to 1 allows the comparison at different frequencies.”

3 - Paragraph 2 is entitled “switching between strong and weak coupling regimes” but the reported data always shows two nice and well separated polaritonic peaks and the actual light-matter coupling decreases by 25%, not by 100%. I am not sure that is the appropriate title. I would more go for “modulation of the strong coupling regime”

Indeed paragraph 2 has been conceived to illustrate the operating principle of the modulator. For this reason Figure 1c in the paper is a sketch that represents the ideal operation of a modulator. We think it is important to stress here the potential of this solution. However, the referee is right observing that the experimental modulation depth is of the order of 25% and the sample is still in the coupling regime upon application of a bias.

Figure 1. Sample with lower doping ($0.61E12cm^{-2}$): (a) 300K reflectance under different DC bias. (b) Comparison between 300K reflectance under 6V bias (red curve) et under no bias (blue curve).

The choice of the $1.74E12 cm^{-2}$ doped sample has been a thoughtful one: indeed the less doped sample ($0.6E12 cm^{-2}$) shows a clear switching from the strong to the weak coupling regime as confirmed by Figure 1 above (Figure S3 updated in the Supplementary Material). However, it is inferior to the highly doped sample in terms of modulation contrast, given the frequency range covered by our MIR laser. We therefore preferred to use the $1.74E12 cm^{-2}$ sample.

In order to comply with the referee observation, we have changed the title of Paragraph 2 to:

“Operating principle: modulating the strong coupling leads to reflected beam modulation”

4 - The terminology “ultra-fast” for an object with a cutoff slightly above 1 GHz is in my opinion abused. Ultrafast physics (few optical cycles) happens on timescales well below 200 fs for visible radiation. for Mid-IR this would translate into few ps dynamics, not nanosecond or slightly below....I would suggest to replace the term with something more descriptive of the actual device performance.

We have been tempted for the title by the papers in literature (Ref. 16 for example). In any case the referee is right in pointing this out, and the title has been changed from ultra-fast to just fast:

“Fast amplitude modulation of mid-IR free-space beams at room-temperature up to 1.5GHz”

The term “ultra-fast” has been changed to “fast ” all along the paper, when referred to our sample.

Minor/technical remarks

-It would be worth citing, in the same spirit of Ref.15, also Jessop et al., Applied Physics Letters, 108 (2016)

-Figure 1 (b) would benefit from a x-y axis schematic

-Figure 2(c): wouldn't be better to report also the contrast, as this is the quantity mainly discussed in the text?

-Figure 4 (a): I would use a dB vertical scale, so the -3 dB cutoff would be immediately evident.

We have addressed all these technical remarks in the revised version apart from the dB scale: we preferred to keep it in log-log.

-The modulator operates in reflection geometry, that for some application is not ideal. Do the Authors have prospects for implementing such concepts in a transmissive device?

The reviewer is right, in general modulators operate in a transmission geometry. Implementing such a configuration means a significant effort in terms of clean-room processing, essentially because we should perform lithography on both sides of the structure. For the moment we are focused on the optimization of the active region, in order to achieve ultra-fast modulation speeds (several GHz) in reflection. For this, we are moving to a coupled quantum well system, plus a geometry with coplanar access lines, as reported in the conclusions. Once that goal is achieved, it will become useful for applications to invest effort in implementing a transmission geometry.

As per applications, there is a compromise solution even in a reflection geometry that increase practicality: operating the device at 45deg, we can have the input and output beams at at 90 degrees angle.

I have a suggestion for an experiment that could shed more light into the modulator's behavior: the reflected/modulated signal could be filtered with an FTIR and then detected with a (fast) lock-in, in the same spirit as the SWIFTs spectroscopy done on QCL combs. In this way it could be assessed the Mid-IR spectral content that is

modulated. I understand that it is a quite involved measurement and it is not strictly required for the present paper but it could be valuable for future works.

We thank the reviewer for this interesting and valuable suggestion. Indeed SWIFT measurements have proven very powerful in the context of mid-IR combs, and it would be interesting to implement the same concept for modulators.

The measurement requires a certain amount of equipment and setup modifications, and we are starting to think about it thanks to the reviewer comment.

Indeed a control on the MIR spectrum of the modulated beam can improve our understanding of the modulator's behavior.

Reviewer #2 (Remarks to the Author):

This paper demonstrates a mid-infrared intensity modulator for modulating beams in free space at room temperature, with an operation speed up to 1.5 GHz.

.....

The results presented show a breakthrough in terms of speed compared to the reported works. However, it is difficult to be considered as a significant achievement. The working principle is similar to Refs 15 and 16. The major difference is that the way of tuning the ISB transitions is replaced by depleting the QWs using a Schottky contact, which is applied in Ref. 13.

As the referee points out, the “results presented [in our work] show a breakthrough in terms of speed compared to the reported works”. In our opinion this is the significant achievement, and the fact that the operating principle bears some similarity to previous work does not remove impact from the current demonstration.

As a matter of fact, Refs. 15 (Benz et al.) and 13 do not report absolutely any measurement about the device speed. And Ref. 16 (Lee et al.) does not report any spectrum analyzer “beatnote”, nor any device frequency bandwidth. Without these crucial figures of merit, nothing can be concluded about the device fast response.

Our work is the first one in the literature that presents these (ultra)-fast characterizations typical of RF devices, widely accepted in the RF community, and beyond 1 GHz, on a compact mid-IR modulator. This adds important credibility to our results.

We also wish to highlight that modulating the charge instead of using the Stark effect (as in Ref. 16) is not a minor difference. A diagonal transition is necessary for the Stark effect, hence the corresponding oscillator strength is reduced and it is necessary to dope more to achieve the same Rabi splitting. Structures with more doping require larger bias to achieve similar frequency tuning, and this can be a problem when targeting device performance. We observe in fact that, since Ref. 16 publication in 2014 using Stark effect, there has been no follow-up from the same team.

Based on this I also disagree with the claim in the conclusion that this is possibly the first report of practical device relying on the strong light-matter coupling.

We agree with the referee comment, and we modified the phrase as follows:

“The device operates by modulating the strong coupling regime at fast rates, one of the few demonstrations of a practical device relying on the strong light-matter coupling regime.”

Because of the similar principle, the breakthrough in the speed is mainly contributed by the smaller device dimension that reduces the RC time constant. E.g. the larger device in this work with a surface area of $5 \times 10^4 \mu\text{m}^2$, has a cut off frequency at 150 MHz. Ref 16 achieves a 10 ns response time (pulsar source limited) with a $16 \times 10^4 \mu\text{m}^2$ surface area, which is comparable.

This is true, and it is very clearly explained in the text (see pages 6/7 revised version). We obtained the 1.5 GHz speed modulation on the smaller devices.

Besides, I would like to point out some issues I found in the paper:

1. The paper is mostly well written, but overall it is so concise that necessary details are missing to provide a comprehensive understanding. For example:

a. Section 2 is too short to clearly introduce the working principle of the device.

We expanded section 2 as follows, in order to make it clearer:

“Let’s consider a laser tuned to the bare cavity frequency $\nu_{las} = \nu_{cav}$ impinging on the device: almost all the intensity is reflected back since $R^{NoBias}_{\nu_{las}} \sim 1$. By applying a bias to the structure we can effectively empty an arbitrary number of wells and finally change the coupling condition between the cavity mode and the ISB transition. In the best case scenario - the ideal device - we can induce a transition to the weak coupling regime. The corresponding reflectivity spectrum is sketched in Figure 1(c) (solid orange line): only the bare cavity transition is visible, as polaritons are no more the eigenstates of the system. At the laser frequency $\nu_{las} = \nu_{cav}$ we have $R^{WBias}_{\nu_{las}} \ll R^{NoBias}_{\nu_{las}}$: the reflected laser beam is amplitude modulated with an elevated contrast”

b. Insets of Fig. 1(c) and the symbols are not detailed.

c. There are too little comments on Fig. 3(b), making it difficult to understand what the data represent

d. There’s no definition for the ‘normalized response’ in Fig. 4(a)

e. No introduction to ‘sideband/carrier ratio’

Points b-e have been addressed.

2. The theoretical Rabi splitting example shown in Fig. 1(c) is too ideal compared to the practical device. The two polaritons are too separated to leave a long region of $R \sim 1$ (may refer to Fig.1 in DOI: 10.1039/c7nr06917k). In the introduction, it is better to be introduced as a significant drop on the reflectance when switching from the strong coupling regime to the weak coupling regime, rather than saying $R=1$ to $R=0$, as we only have 10% change in the experiment. It would also be better to change the x-axis to frequency to be consistent with the descriptions in the caption and the main text.

Indeed paragraph 2 has been conceived to illustrate the operating principle of the modulator. For this reason, Figure 1c in the paper is a sketch that represents the ideal operation of a modulator. We think it is important to stress here the potential of this solution. However, the referee is right, the current version of Fig. 1c is too idealized. The figure has been modified in order to make it more realistic.

Please also see the answer to point 3 of first reviewer that justifies the choice for a sample where we modulate the strong coupling regime, instead of clearly moving out to the weak coupling regime.

3. Both Fig. 1(c) and Fig. 2(a) are better to be plotted without an offset to directly observe the reflectance change. +6V and -6V of Fig. 2(a) can be plotted separately but compared with the zero bias. Meanwhile, the reflectance values are important for Fig. 2(a), which provides the information of how much the reflectance change and the insertion loss. These modifications also facilitate correlating them to Fig.2(c).

Figure 1(c) and Figure 2(a) has been modified following the reviewer's suggestions.

In Figure 2(a) only the +6V curve has been left since without offset the -6V curve is almost completely super-imposed to the +6V one.

4. It's better to label the surface area or the length of each boundary in Fig. 2(b), pointing out which is the bonding pad and which is the wire gating area.

We have introduced this information in the figure

5. y-label of Fig. 2(c) can be changed to modulation depth since it has been defined in the main text. Too difficult to read the equation in the label.

The correction has been introduced

6. The functions of some components in Fig. 5(a) are not introduced, e.g. PM, Vis-Cam, WL and MIR Cam. If they are unnecessary to be introduced they can be removed from the diagram. Otherwise some simple introductions are needed.

We have clarified the roles of these components.

7. Except the speed difference, are there any differences in the performance between the smaller and the larger devices?

No we didn't measure major differences between large and small devices (for example in reflectance under DC bias), since both the active region and the fabrication process are the same.

8. Some typos:

'Ghz', and the missing of the frequency number (The QC laser frequency is ...) in the caption of Fig. 3.

The paragraph below Fig. 4, 'a typical 100x100 μm^2 '. Should it be $2 \times 10^4 \mu\text{m}^2$?

The typos have been corrected

Reviewer #3 (Remarks to the Author):

In this manuscript, the authors present a novel type of mid-infrared modulator surface based upon an intersubband polariton. While modulation of infrared light by intersubband transitions has been performed before numerous times, **the use of a polaritonic transition to do so is relatively new, and no other report of modulation above 1 GHz has been reported. Hence the demonstration is novel, and the importance is potentially high, since the mid-infrared lacks for high-speed modulator components that are separate from a QCL.** The results are convincing, particularly where the reflectance spectrum response measured by FTIR is matched with the modulation strength measured using a tunable laser.

Therefore, I recommend publication provided several points can be addressed :

1 - I am confused by the spectra in Fig. 3(b). It appears there is a modulated signal even when the mid-IR laser is not on? Can the authors explain this? Is it related to modulation of reflected background radiation by the modulator? If so, why is it not present for modulation at 100 MHz, but it is present for the other 3 modulation frequencies? Why is the noise background higher for when the laser is on?

See also answer 2, Referee 1.

The residual signal we have on the spectrum analyzer (SA), even when the MIR laser is off, is pure RF noise coming from the measurement circuit. In fact we keep the RF synthesizer on during the measurement of the red curve with laser off, and there is evidently a small cross talk between the synthesizer and the spectrum analyzer. This often happens in RF setups. The red curves are essentially our noise floors.

It is not present at 100 MHz as the cross talk effect is more and more effective towards GHz-level frequencies.

2 - Given that Lee et al (Adv. Opt. Mat. 10.1002/adom.20140018) previously demonstrated a high-speed polaritonic mid-IR modulating metasurface, the statement in the conclusion that this is the “possibly the first report of a practical device relying on the strong light-matter coupling regime” is not warranted.

We agree with the referee comment, and we modified the phrase as follows:

“The device operates by modulating the strong coupling regime at fast rates, one of the few demonstrations of a practical device relying on the strong light-matter coupling regime.”

3 - What about modulation at voltages greater than 6 V? Can the polariton be completely switched off?

We preferred to fix the maximum bias at the Schottky opening voltage to avoid current flowing into the device. The motivation is that we don't want to stress our device thermally. However, as soon as current flows through the device, in general electric gating becomes ineffective. We therefore do not expect that polaritons can be completely switched off in this

sample. It is instead possible in a lower doped sample. See Supplementary Material (revised version), and answer 3 to referee 1.

4 - In general the preparation and proofreading is a little sloppy.

o The Supplementary has many typos and misspellings and there appear to be missing references in it.

We have corrected the typos.

In the main text I noted:

o Fig 4. Caption – no units are given for area.

o Misspelling: “synthesizer”:, “bandwidth”, “Ghz”, “stark” (should be capitalized),

o Italicization is haphazard – some variables are not italicized and should be, some units are italicized and should not be.

These issues have been corrected.

Reviewers' Comments:

Reviewer #1:

Remarks to the Author:

The Authors answered satisfactorily to my concerns and modified the paper accordingly.
The paper now can be accepted for publication.

Reviewer #2:

Remarks to the Author:

The authors have well addressed most of the questions and the paper has been improved.

Regarding my major concern on the novelty of the device, the authors have explained in the response letter, however it should also be highlighted in the manuscript. E.g. an brief introduction to the different mechanism & advantages of the proposed device compared to the reported works, especially Ref 13-16, can be given after the last paragraph of the Introduction section.

Other issues: add introduction to the abbreviations AM and FM.

Reviewer #3:

Remarks to the Author:

In my opinion, the authors have addressed the comments well, and I recommend publication.